# How Is Leisure Related to Wellbeing and to Substance Use? The Probable Key Role of Autonomy and Supervision

**DOI:** 10.3390/children10050773

**Published:** 2023-04-25

**Authors:** Gina Tomé, Fábio Botelho Guedes, Ana Cerqueira, Catarina Noronha, Joaquim Castro de Freitas, Teresa Freire, Margarida Gaspar de Matos

**Affiliations:** 1Institute of Environmental Health (ISAMB), Aventura Social, Faculty of Medicine, University of Lisbon (FMUL), 1649-028 Lisbon, Portugalcatarinanoronha26@live.com.pt (C.N.); 2Faculty of Human Kinetics, University of Lisbon (FMH-UL), 1495-751 Lisbon, Portugal; 3Rede de Informação aos Jovens–Eurodesk Portugal, 4715-558 Braga, Portugal; 4School of Psychology, Psychology Research Centre, University of Minho, 4710-057 Braga, Portugal; 5APPSYci—Applied Psychology Research Center Capabilities & Inclusion, ISPA–University Institute, 1100-304 Lisbon, Portugal

**Keywords:** leisure activities, well-being, substance use, adolescents, autonomy, peer pressure, adult monitoring

## Abstract

The present research is based on a large and representative national survey and intends to analyse the correlation of several leisure activities with risk, and with health and well-being outcomes. This work is part of the Health Behaviour in School-aged Children (HBSC) study, a collaborative WHO international study that aims to explore the school-aged children behaviour regarding health and risk behaviours in their life contexts. Participants were 8215 Portuguese adolescents, randomly chosen from those attending the 6th, 8th, 10th and 12th grades in 2018. The sample included 52.7% of girls and the mean age was 14.36 years old. Descriptive and comparative analyses were performed (ANOVAS and Chi-Square). The results of the present study suggested that several leisure activities, namely sports and social engagement activities (politic involvement and participation, religious activities, scouting and volunteer work), are associated with the adolescents’ well-being and life satisfaction. However, these types of activities can also be associated with an increase in substance use. However, some activities are also associated with risky behaviour. Identifying activities that promote well-being in young people can be important for professionals, families and public policies.

## 1. Introduction

The participation in social groups and/or structured activities (e.g., scouts, volunteering, sports, religious and artistic groups) can be a protective factor for the positive and healthy development of the adolescents [1]. Leisure is important, but as leisure time is often associated with peer-to-peer interaction in absence of adult monitoring, an effort has to be made in order to introduce this autonomy step by step, decreasing the odds of it becoming associated with risk-taking in absence of adult monitoring, or providing, at least in the younger children, a gradient regarding adult monitoring [2]. The present research is based on a large and representative national survey and aims to analyse the association between being involved in different types of leisure activities, namely sports, reading, and social engagement activities (politic involvement and participation, religious activities, scouting and volunteer work), and adolescents’ well-being and life satisfaction or on the other hand substance use. The influence of gender and age was also addressed.

### 1.1. Leisure Activities and Wellbeing

A study by Trainor et al. (2010) demonstrated a relationship between the leisure activities and the adolescents’ psychological wellbeing, as individuals with a higher level of wellbeing tend to get involved in more structured activities [3]. There is evidence to suggest that more structured and organized activities are reflected in more positive results regarding the adolescents’ development [4,5,6]. These authors highlight the positive effects of physical activity on the mental and physical health of the adolescents, which contributes to their psychosocial wellbeing and quality of life. Participation in civic and community activities plays an important role with regard to the adolescents’ positive development. Its effects are reflected in an increased academic performance and psychosocial wellbeing as well in the reduction in their involvement in risk behaviours. Religious organizations are an example of one of the contexts that can contribute to the adolescents’ involvement in activities that promote civic and community participation, which in turn contributes to their psychosocial development and wellbeing [7,8,9].

Participation in activities such as scouting represents an opportunity for the adolescents to learn and to develop skills through non-formal education. The evidence in the literature points to the benefits that the structured and voluntary activities can have regarding the positive development of the adolescents [10,11,12,13,14].

Health, wellbeing, and psychosocial development are factors that can be influenced by engaging in leisure activities that are relevant to the adolescents and that serve a particular purpose. In fact, the lack of meaningful activities can contribute to the increased likelihood of engaging in risk behaviours or developing health-related problems, such as increased substance use and difficulties in interpersonal relationships, among others [10,15].

### 1.2. Leisure Activities and Protective Factors

Participation in leisure activities is seen as a protective factor against substance abuse and risk behaviour as well as a predictor of the adolescents’ psychological wellbeing [11,16,17,18]. The literature points to a relationship between structured out-of-school activities and increase in substance use (i.e., alcohol, tobacco, and marijuana), namely those activities that involve spending time with friends without adult supervision [19]. For several reasons, the peer group contributes to the adolescents’ positive development [20], but it can also represent a risk factor for the involvement in risk behaviours [21,22]. In the same way, and as an example, although sport can function as a protective factor against substance use, this activity can also appear associated with an increase in the adolescents’ alcohol use [2,19,23,24].

The literature proposes that the activities that are less structured and often unsupervised (e.g., watching television, reading or playing videogames) are more likely to promote the adolescents’ involvement in risk behaviours [3,16,25,26]. It is also proposed that the activities that involve the participation of peers are more likely to promote behaviours such as smoking, drinking alcohol or using drugs [3], while activities that require adult supervision and participation tend to be more discouraging of such behaviours [25].

A systematic review by Lisha and Sussman (2010) examined the relationship between participation in sports activities and drug use in adolescents [27]. The results point in the following direction: (1) recreational physical activity is associated with higher levels of alcohol use and binge drinking [28,29]; (2) athletes engage more frequently in alcohol-related risk behaviours in comparison to non-athletes; (3) athletes who practice in a formal context and with a frequency between three and five times per week are more likely to report alcohol use; (4) athletes present a higher predisposition to engage in risk behaviours and lifestyle patterns associated to alcohol and drug consumption [30]; (5) a perceived athletic identity is correlated with alcohol consumption [31]. Thus, this review reported a generally positive association between alcohol use and participation in sports [27]. With regard to gender differences, the study revealed that (1) male hockey and female soccer players are the most likely to report high levels of alcohol consumption, while cross-country/track athletes report the lowest levels; (2) sport participation is associated with increased alcohol consumption among female students but is unrelated to drinking practices among males; (3) regular practice of individual sport or a team sport is correlated positively with repeated use of alcohol for both sexes; (4) a sex difference was detected where regular practice of sport is correlated with recent drunkenness for males only, and repeated use of alcohol for females only [27].

As participation in unstructured leisure activities can also be associated with risk behaviours such as substance use [32,33,34] and delinquency [20,21,35,36,37], these types of activities can also contribute to behavioural problems that, in turn, can lead to unhealthy developmental outcomes [33]. Thus, it is important to consider the type of activities in which the adolescents are involved (i.e., structured and supervised vs. non-structured and unsupervised), since these can function as protective or risk factors with regard to young people’s positive development [38].

### 1.3. Portuguese Youth

In line with Portuguese youth-related public policies, the Portuguese Institute of Sports and Youth, I.P. (IPDJ) has the mission of executing an integrated and decentralized policy for the areas of sport and youth, in close collaboration with public and private entities, namely with sports organizations, youth associations, students and local authorities. In the past years, several programmes were launched to promote health and healthy lifestyles for young people. One of the most relevant is “Cuida-te+” (“Take care of yourself”), a program aimed at promoting health and healthy lifestyles for young people between 12 and 25 years of age as the final target population. In addition to that, a strategic target population is working as intermediate actors that have a potentially influential role in promoting behaviours beneficial to the health of young people, namely health professionals, youth workers, community intervention professionals, leaders of youth associations and their federations and families, and other young people as stakeholders. Policies, training materials, promotion campaigns and online support are key components providing support for young people and organizations in promoting wellbeing and healthy behaviours. At the European level, this is also a priority. In 2018, a resolution was approved of the Council of the European Union and the Representatives of the Governments of the Member States meeting within the Council on a framework for European cooperation in the youth field: The European Union Youth Strategy 2019–2027 (2018/C 456/01). This important resolution provides the legal background to build the “Youth Strategy 19–27” along all the EU countries as well as the European Institutions. This Strategy has 11 youth goals. The Youth Goal number 5 is related to mental health and wellbeing. The background information shows that “A significant and increasing number of young people across Europe are expressing their concern at the prevalence of mental health issues such as high stress, anxiety, depression and other mental illnesses amongst their peers. Young people cite the immense societal pressures they face today, and express a need for better youth mental health provision”. The goal is to, within the defined timeline, “Achieve better mental wellbeing and end stigmatization of mental health issues, thus promoting social inclusion of all young people”. This document should have an impact in the national policies and practices, as it is implemented together with organizations, associations and institutions working in the Youth Sector framework.

The present research is based on a large and representative national survey and aims to analyse the association between being involved in different types of leisure activities, namely sports, reading, and social engagement activities (politic involvement and participation, religious activities, scouting and volunteer work), and adolescents’ wellbeing and life satisfaction or on the other hand substance use. The influence of gender and age was also addressed.

## 2. Materials and Methods

This work is part of the Health Behaviour in School-aged Children (HBSC) study [23,24,39]. The HBSC is a collaborative WHO study, undertaken in 44 countries and aiming to study the school-aged children behaviour regarding health and risk behaviours. Portugal has been part of this group of countries since 1996. The HBSC is a school-based survey of the adolescents’ health behaviours, carried out every 4 years. Collected data are used at a national and international level, using an internationally standardized methodological protocol [39] that, in general, intends to (1) gain a new insight into the adolescents’ health and wellbeing, (2) understand the social and psychological determinants of health and (3) incorporate policies to improve the adolescents’ lives. For the purpose of this study, several variables were analysed regarding to socio-demographics, leisure activities, wellbeing, life satisfaction and risk behaviours.

### 2.1. Participants

The 2018 study provided national representative data of 8215 Portuguese adolescents, randomly chosen from those attending the 6th, 8th (middle school), 10th and 12th grades (high school) during the 2017/2018 academic year. The sample included 52.7% of girls and 47.3% of boys, whose mean age was 14.36 years old (standard deviation 2.28). The present study used a subset of the 8th (*n* = 2766), 10th (*n* = 1711) and 12th graders (*n* = 1218) to represent middle school and high school. These students were randomly selected from 42 public school groups, in a total of 476 classes and in a national sample geographically stratified by Education Regional Divisions in Portugal. The overall procedure has been described elsewhere (24;39); in brief, this study has the approval of a scientific committee, an ethical national committee and the national commission for data protection and strictly follows all the guidelines for protection of the human rights; the adolescents’ participation in the survey was voluntary and anonymous. The sample was nationally representative of the respective grade levels, according to information on the number of students in public schools, from the general directorate of education.

### 2.2. Measures and Variables

The variables under analysis in the present study evaluate the type of leisure activities, the levels of wellbeing and the involvement in substance use of Portuguese adolescents. These variables are presented in Table 1.

Leisure activities were evaluated using a variable measured by one single item about activities they partake in in free time, with a list of different options of activities to choose and answers showing how often these activities are performed (Practicing a sport; Reading; Volunteer activities; Religious activities; Scouting activities; Political activities). All activities have four answer options: 1—Rarely or never; 2—Only on the weekend; 3—Every day or almost every day; 4—Several hours a day [24,39].

The following scales were used to assess wellbeing: Kidscreeen scale, the global Wellbeing (10-item scale: 1 to 5), with an internal consistency index of a 0.84 [40], and Life Satisfaction scale adapted from Cantril (1965) [41].

Substance use was assessed using the variables for Alcohol and Drug use in the last 30 days [39]. Table 1 shows details on these measures. children-10-00773-t001_Table 1Table 1Measures and variables under study.Variables/InstrumentsMeasureLeisure activitiesVariable measured by one single item with the question “What do you usually do in your free time?” with six options of activities: Practice a sport; Read; Volunteer activities; Religious activities; Scouting activities; Political activities. All activities have four answer options: 1—Rarely or never; 2—Only on the weekend; 3—Every day or almost every day; 4—Several hours a day.WellbeingGlobal health scale with ten items (Kids-10, from the original version of the Kidscreen-52 scale). Kids-10 contains five answer options: 1—Never; 2—Rarely; 3—Sometimes; 4—Often and 5—Always. Higher values correspond to higher wellbeing.Life satisfactionScale adapted from Cantril (1965) [41], consisting of 11 steps: the top of the ladder “10” is the best possible life and the bottom “0” is the worst possible life. Substance use
Alcohol use in the last 30 daysWith seven response options, 1—Never and 7–30 days (or more).Tobacco use in the last 30 daysWith seven response options, 1—Never and 7–30 days (or more).Drug use in the last 30 daysWith seven response options, 1—Never and 7–30 days (or more).


The data were analysed using the Statistical Package for Social Sciences (SPSS) version 24 for Windows. Descriptive and comparative analyses were performed (ANOVAS and Chi-Square) for the variables under study.

## 3. Results

In the first step, the analysis aimed to understand the association of gender, age, grade and the preference with different types of leisure activities, and in the second step, we analysed the association of the different types of leisure activities with wellbeing and life satisfaction.

### 3.1. Types of Leisure Activities and Gender and Grade

#### 3.1.1. Gender

It was determined by Chi-square that boys participated more in sport activities (χ^2^ = 72.577(1), *p* ≤ 0.000, 74.5%) than in scouting activities (χ^2^ = 11.942(1), *p* ≤ 0.01, 16.9%) or in political activities (χ^2^ = 20.961(1), *p* ≤ 0.000, 15.6%). In turn, girls read more (χ^2^ = 162.553(1), *p* ≤ 0.000, 57.6%) and participated more in religious activities (χ^2^ = 8.048(1), *p* ≤ 0.01, 33.4%). The results for volunteering activities were not statistically significant (see Table 2).

#### 3.1.2. Grade

It was determined by Chi-square that the 8th grade adolescents participated more in sport activities (χ^2^ = 35.436(2), *p* ≤ 0.000, 72.3%) and read more (χ^2^ = 15.509(2), *p* ≤ 0.000, 52%) in comparison to volunteering activities (χ^2^ = 40.260(2), *p* ≤ 0.000, 22.6%), religious activities (χ^2^ = 92.546(2), *p* ≤ 0.000, 38.7%), scouting activities (χ^2^ = 84.573(2), *p* ≤ 0.000, 20.2%) and political activities (χ^2^ = 56.609(2), *p* ≤ 0.000, 17.3%) (see Table 3).

### 3.2. Types of Substance Use and Leisure Activity (All Activities Recoded “Yes” or “No”)

Table 4 shows the differences in types of risk behaviours (in terms of consumption) according to the type of leisure activity.

It was determined that the adolescents who practice more sports reported a lower tobacco consumption in the last 30 days (fewer days) (χ^2^ = 6.694(2), *p* ≤ 0.05, 93.2%). The results for sport activities and marijuana and alcohol consumption were not statistically significant.

The adolescents who engage in reading activities reported a lower alcohol consumption in the last 30 days (fewer days) (χ^2^ = 11.832(2), *p* ≤ 0.05, 86.9%). The results for reading activities and marijuana and tobacco consumption were not statistically significant.

Engaging in volunteering activities was associated with a higher consumption of alcohol in the last 30 days (more days) (χ^2^ = 9.230(2), *p* ≤ 0.01, 4.5%). The results for volunteering activities and marijuana and tobacco consumption were not statistically significant.

The adolescents who are involved in religious activities showed a lower consumption of tobacco (χ^2^ = 8.705(2), *p* ≤ 0.05, 94.1%) and alcohol (χ^2^ = 11.181(2), *p* ≤ 0.01, 87.6%) in the last 30 days (fewer days). The results regarding the religious activities and the marijuana consumption were not statistically significant.

In regard to involvement in scouting activities, an association with higher marijuana and alcohol consumption was revealed; the adolescents who engage in these activities consumed more marijuana (χ^2^ = 15.077(2), *p* ≤ 0.01, 2.6%) and more alcohol (χ^2^ = 7.427(2), *p* ≤ 0.05, 4.6%) in the last 30 days (more days). The results for scouting activities and tobacco consumption were not statistically significant.

Finally, it was observed that the adolescents who participate in political activities showed a higher marijuana (χ^2^ = 15.778(2), *p* ≤ 0.001, 2.4%), tobacco (χ^2^ = 14.439(2), *p* ≤ 0.01, 6.8%) and alcohol consumption (χ^2^ = 21.588(2), *p* ≤ 0.000, 5.7%) in the last 30 days (more days).

### 3.3. Adolescents’ Life Satisfaction and Wellbeing, and Participation in Each Type of Activity

Table 5 shows the differences in the adolescents’ life satisfaction and wellbeing according to the type of leisure activity in which they are involved. The adolescents who practice sport have higher life satisfaction (M = 7.4, SD = 1.6), F(1,4413) = 77.446, *p* ≤ 0.000 and wellbeing (M = 37.6, SD = 6.7), F(1,4413) = 266.807, *p* ≤ 0.000. The adolescents who engage in volunteer activities have higher life satisfaction (M = 7.5, SD = 1.8), F(1,4403) = 10.003, *p* ≤ 0.001 and lower wellbeing (M = 35.8, SD = 7.8), F(1,4403) = 7.818, *p* ≤ 0.01. Regarding the religious activities, it was observed that the adolescents who engage in this type of activity have higher life satisfaction (M = 7.4, SD = 1.7), F(1,4393) = 10.842, *p* ≤ 0.001. The results for wellbeing were not statistically significant. Regarding scouting activities, the results showed that adolescents who engage in this activity have higher life satisfaction (M = 7.5, SD = 1.9), F(1,4414) = 11.462, *p* ≤ 0.001 but lower wellbeing (M = 35.5, SD = 7.9), F(1,4414) = 14.853, *p* ≤ 0.000. Finally, the adolescents who are involved in political activities have higher life satisfaction (M = 7.5, SD = 1.9), F(1,4412) = 6.374, *p* ≤ 0.05 but lower Wellbeing (M = 34.9, SD = 7), F(1,4412) = 29.545, *p* ≤ 0.000. The results of reading activities were not statistically significant.

## 4. Discussion

Our results have suggested that several leisure activities, namely sports and social engagement activities (politic involvement and participation, religious activities, scouting and volunteer work), are in general associated to the adolescents’ higher levels of wellbeing and life satisfaction. However, it is important to note that a few of them also appear associated to increased substance use [27,42,43]. Often, the adolescents are expected to balance a multitude of obligations, such as working out, having a social life, attending classes, studying and engaging in leisure activities.

A study by Trainor et al. (2010) demonstrated a relationship between the leisure activities and the adolescents’ psychological wellbeing [3]. However, the literature points to a relationship between the out-of-school activities and the increase in substance use (i.e., alcohol, tobacco, and marijuana), namely those activities that involve spending time with friends without adult supervision [19].

The present results have suggested that participation in leisure activities tends to be associated with protective health and wellbeing conditions, as well as associated to substance use [32,33,34]. Especially the practice of sport is an activity that appears as a protective factor for risky behaviour and for the promotion of wellbeing in young people. However, according to the results obtained, activities such as volunteering do not demonstrate to be protective activities, as expected, in line with activities described in several studies as associated with risky behaviour [20,21,32,33,34].

Nevertheless, it is important to consider the type of activities in which the adolescents are involved (i.e., structured and supervised versus non-structured and unsupervised), since these can function as protective or risk factors with regard to the adolescents’ positive development and adjustment [38].

The peer relationships and the adolescents’ autonomy can be essential for a positive development during adolescence, but they are also related with more exploration behaviours related to the adolescent period, such as trying to learn about themselves or others or circumstances that can lead to the emergence of risk behaviours, namely substance use. There is an increased risk when the adolescents, especially the younger ones, are not adequately supervised, when the leisure activities provide extra stress and fatigue, or when there is peer pressure leading to health-undermining decisions.

When designing out-of-school leisure activities, providing either structured activities or opportunities to unstructured activities, it is important to ensure that these activities will not contribute to an abusive increase in stress or fatigue, as well as to an abusive unsupervised contact with peers, because unsupervised contact with peers is one of the features associated with risk behaviours in adolescents, especially for younger adolescents, because mechanisms of cognitive, emotional and social development and of autonomy are emerging [2].

## 5. Limitations

The types of variables included in the study limit statistical analysis, making it difficult to carry out more robust analyses. It will be important in future studies to deepen the association between leisure activities and risky behaviours and low level of life satisfaction among young people who practice some leisure activities.

## 6. Conclusions

The results suggest that the involvement of young people in structured and monitored leisure activities promotes adolescent wellbeing. However, when designing leisure activities, it is important to provide either structured activities or opportunities to unstructured activities and ensure that these activities will not contribute to increase in stress or fatigue, as well as to a negative unsupervised contact with peers, because unsupervised contact with peers is one of the features associated with risk behaviours in adolescents [2].

An important cue for future researches is to estimate the importance of ensuring that leisure activities provide a balance in terms of the adolescents’ autonomy and adult monitoring (i.e., not providing excessive autonomy without adult monitoring, which can result in negative consequences). Another important cue is to estimate whether it makes a difference regarding wellbeing and health-related behaviours when the leisure activities are tailored to the interests of each adolescent in order to become meaningful and not only a way of spending time/wasting time. Finally, it is important to determine whether the opportunity of engaging in leisure activities is associated to excessive time, excessive competition or anyhow excessive additional stress.

## Figures and Tables

**Table 2 children-10-00773-t002:** Participation in leisure activities by gender.

Gender								
**Activities**		**Boy**		**Girl**		**Total**	**χ^2^**	**df**
		**N**	**%**	**N**	**%**			
**Sport**	No	505	**25.5**	912	**37.5**	1417	72.577 ***	1
	Yes	1478	**74.5**	1520	**62.5**	2998		
**Read**	No	1229	**61.9**	1037	**42.4**	2266	162.553 ***	1
	Yes	766	**38.4**	1410	**57.6**	2176		
**Volunteering**	No	1616	81.5	1972	81.4	3588	0.004	1
	Yes	367	18.5	450	18.6	817		
**Religious activities**	No	1400	**70.6**	1608	**66.6**	3008	8.048 **	1
	Yes	582	**29.4**	805	**33.4**	1387		
**Scouting**	No	1653	**83.1**	2109	**86.9**	3762	11.942 **	1
	Yes	335	**16.9**	319	**13.1**	654		
**Political Activity**	No	1675	**84.4**	2163	**89**	3838	20.961 ***	1
	Yes	310	**15.6**	266	**11**	576		

Adjusted residual values ≤ 1.9 are bold. *** *p* ≤ 0.000; ** *p* ≤ 0.01; df = freedom degrees.

**Table 3 children-10-00773-t003:** Participation in leisure activities by grade.

Grade										
**Activities**		**8th**		**10th**		**12th**		**Total**	**χ^2^**	**df**
		**N**	**%**	**N**	**%**	**N**	**%**			
**Sport**	No	546	**27.7**	476	33.9	395	**37.9**	1417	35.436 ***	2
	Yes	1422	**72.3**	930	66.1	646	**62.1**	2998		
**Read**	No	954	**48**	772	**54.8**	540	51.7	2266	15.509 ***	2
	Yes	1034	**52**	637	**45.2**	505	48.3	2176		
**Volunteering**	No	1519	**77.4**	1175	**83.7**	894	**86**	3588	40.260 ***	2
	Yes	443	**22.6**	229	**16.3**	145	**14**	817		
**Religious activities**	No	1199	**61.3**	1008	**71.8**	801	**77.4**	3008	92.546 ***	2
	Yes	758	**38.7**	395	**28.2**	234	**22.6**	1387		
**Scouting**	No	1570	**79.8**	1246	**88.4**	946	**91**	3762	84.573 ***	2
	Yes	397	**20.2**	164	**11.6**	93	**9**	654		
**Political Activity**	No	1625	**82.7**	1279	**90.6**	934	**90.1**	3838	56.609 ***	2
	Yes	340	**17.3**	133	**9.4**	103	**9.9**	576		

Adjusted residual values ≤ 1.9 are bold; *** *p* ≤ 0.000; df = freedom degrees.

**Table 4 children-10-00773-t004:** Leisure activities and substance use.

Sport								
**Substance Use**		**No**		**Yes**		**Total**	**χ^2^**	**df**
		**N**	**%**	**N**	**%**			
**Marijuana/last 30 days**	Never/1–2 days	1364	96.3	2918	97.3	4282	3.850	2
	Few days	32	2.3	50	1.7	82		
	Many days	21	1.5	30	1	51		
**Tobacco/last 30 days**	Never/1–2 days	1289	**91**	2793	**93.2**	4082	6.694 *	2
	Few days	56	4	92	3.1	148		
	Many days	72	**5.1**	113	**3.8**	185		
**Alcohol/last 30 days**	Never/1–2 days	1220	86.1	2535	84.6	3755	1.832	2
	Few days	157	29.7	372	12.4	529		
	Many days	40	30.5	91	69.5	131		
**Read**								
**Substance Use**		**No**		**Yes**		**Total**	**χ^2^**	**df**
		**N**	**%**	**N**	**%**			
**Marijuana/last 30 days**	Never/1–2 days	2196	96.9	2112	97.1	4308	0.507	2
	Few days	41	1.8	41	1.9	82		
	Many days	29	1.3	23	1.1	52		
**Tobacco/last 30 days**	Never/1–2 days	2085	92	2024	93	4109	2.350	2
	Few days	76	3.4	71	3.3	147		
	Many days	105	4.6	81	3.7	186		
**Alcohol/last 30 days**	Never/1–2 days	1888	**83.3**	1892	**86.9**	3780	11.832 **	2
	Few days	306	**13.5**	225	**10.3**	531		
	Many days	72	3.2	59	2.7	131		
**Volunteering**								
**Substance Use**		**No**		**Yes**		**Total**	**χ^2^**	**df**
		**N**	**%**	**N**	**%**			
**Marijuana/last 30 days**	Never/1–2 days	3489	97.2	784	96	4273	5.641	2
	Few days	63	1.8	17	2.1	80		
	Many days	36	1	16	2	52		
**Tobacco/last 30 days**	Never/1–2 days	3332	92.9	742	90.8	4074	4.143	2
	Few days	115	3.2	32	3.9	147		
	Many days	141	3.9	43	5.3	184		
**Alcohol/last 30 days**	Never/1–2 days	3068	85.5	680	83.2	3748	9.230 **	2
	Few days	428	11.9	100	12.2	528		
	Many days	92	**2.6**	37	**4.5**	129		
**Religious Activities**								
**Substance Use**		**No**		**Yes**		**Total**	**χ^2^**	**df**
		**N**	**%**	**N**	**%**			
**Marijuana/last 30 days**	Never/1–2 days	2910	96.7	1352	97.5	4262	2.529	2
	Few days	62	2.1	19	1.4	81		
	Many days	36	1.2	16	1.2	52		
**Tobacco/last 30 days**	Never/1–2 days	2760	**91.8**	1305	**94.1**	4065	8.705 *	2
	Few days	105	3.5	41	3	146		
	Many days	143	**4.8**	41	**3**	184		
**Alcohol/last 30 days**	Never/1–2 days	2525	**83.9**	1215	**87.6**	3740	11.181 **	2
	Few days	393	**13.1**	133	**9.6**	526		
	Many days	90	3	39	2.8	129		
**Scouting**								
**Substance Use**		**No**		**Yes**		**Total**	**χ^2^**	**df**
		**N**	**%**	**N**	**%**			
**Marijuana/last 30 days**	Never/1–2 days	3662	**97.3**	621	**95**	4283	15.077 **	2
	Few days	65	1.7	16	2.4	81		
	Many days	35	**0.9**	17	**2.6**	52		
**Tobacco/last 30 days**	Never/1–2 days	3492	92.8	592	90.5	4084	5.530	2
	Few days	124	3.3	24	3.7	148		
	Many days	146	3.9	38	5.8	184		
**Alcohol/last 30 days**	Never/1–2 days	3206	85.2	550	84.1	3756	7.427 *	2
	Few days	456	12.1	74	11.3	530		
	Many days	100	**2.7**	30	**4.6**	130		
**Political Activity**								
**Substance Use**		**No**		**Yes**		**Total**	**χ^2^**	**df**
		**N**	**%**	**N**	**%**			
**Marijuana/last 30 days**	Never/1–2 days	3738	**97.4**	544	**94.4**	4282	15.778 ***	2
	Few days	63	**1.6**	18	**3.1**	81		
	Many days	37	**1**	14	**2.4**	51		
**Tobacco/last 30 days**	Never/1–2 days	3571	**93**	511	**88.7**	4082	14.439 **	2
	Few days	122	3.2	26	4.5	148		
	Many days	145	**3.8**	39	**6.8**	184		
**Alcohol/last 30 days**	Never/1–2 days	3292	**85.8**	462	**80.2**	3754	21.588 ***	2
	Few days	449	11.7	81	14.1	530		
	Many days	97	**2.5**	33	**5.7**	130		

Adjusted residual values ≤ 1.9 are bold; *** *p* ≤ 0.000; ** *p* ≤ 0.01; * *p* ≤ 0.05; df = freedom degrees.

**Table 5 children-10-00773-t005:** Differences in wellbeing and life satisfaction and participation (yes or no) in different types of leisure activities (ANOVA).

Sport	No			Yes				
	**N**	**M**	**SD**	**N**	**M**	**SD**	** *F* **	** *p* **
Life Satisfaction	1417	6.9	1.8	2998	7.4	1.6	77.446	**<0.001**
Wellbeing	1417	33.9	7.5	2998	37.6	6.7	266.807	**<0.001**
**Read**	**No**			**Yes**				
	**N**	**M**	**SD**	**N**	**M**	**SD**	** *F* **	** *p* **
Life Satisfaction	2266	7.3	1.7	2176	7.3	1.8	0.278	0.278
Wellbeing	2266	36.6	7.1	2176	36.4	7.2	0.774	0.379
**Volunteering**	**No**			**Yes**				
	**N**	**M**	**SD**	**N**	**M**	**SD**	** *F* **	** *p* **
Life Satisfaction	3588	7.3	1.7	817	7.5	1.8	10.003	**0.002**
Wellbeing	3588	36.6	7	817	35.8	7.8	7.818	**0.005**
**Religious Activities**	**No**			**Yes**				
	**N**	**M**	**SD**	**N**	**M**	**SD**	** *F* **	** *p* **
Life Satisfaction	3008	7.2	1.8	1387	7.4	1.7	10.842	<0.001
**Wellbeing**	**3008**	**36.4**	**7.2**	**1387**	**36.6**	**7.3**	**0.455**	**0.500**
**Scouting**	**No**			**Yes**				
	**N**	**M**	**SD**	**N**	**M**	**SD**	** *F* **	** *p* **
Life Satisfaction	3762	7.3	1.7	654	7.5	1.9	11.462	<0.001
Wellbeing	3762	36.6	7.1	654	35.5	7.9	14.853	<0.001
**Political Activity**	**No**			**Yes**				
	**N**	**M**	**SD**	**N**	**M**	**SD**	** *F* **	** *p* **
Life Satisfaction	3838	7.3	1.7	576	7.5	1.9	6.374	**0.012**
Wellbeing	3838	36.7	7	576	34.9	8.3	29.545	<0.001

## Data Availability

Data sharing is not applicable to this article.

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
