# Peer review of "How Is Leisure Related to Wellbeing and to Substance Use? The Probable Key Role of Autonomy and Supervision"

_children, 2023, doi:10.3390/children10050773_

Round 1

Reviewer 1 Report

Dear Authors,

Thank you for your manuscript. Please see my comments below.

Several points should be better specified in the Introduction section. For example, "In fact, the lack of meaningful activities can contribute to the increase likelihood of engaging in risk behaviours or developing health-related problems" (lines 60-61) should be continued specifying the problems.

Also, the study novelty should be better specified.

A paragraph in lines 79-88 (alcohol use in athletes) should be revised regarding gender differences.

In the Methods section, more information should be provided on the KIDSCREEN-10 tool (Cornbach's alpha, reference supporting validity in a national language).

The tables should be more self-explanatory: the meaning of the abbreviation "gl" and *, **, *** should be explained in the footnotes (Tables 2-4).

Next, in Table 5 p-values should be corrected from .000 to < 0.001, .278 to 0.278. Please note, that the MDPI does not support APA styling in numbers.

The discussion section is very scarce. Conclusions are missing and the main study findings are not summarised. Also, study strengths, limitations and practical implications should be provided.

Minor comment. The reference style should be corrected according to MDPI requirements. Reference numbers should be provided in the angle bracket: [2,3], etc.

Author Response

Dear reviewer,
Thank you very much for your very pertinent comments that help us to improve our article.
We have added the requested information in the revised article, which will be submitted shortly.
(lines 60-61) we add the missing information;
Lines 79-88 we add information about the gender difference of the referred study;
In the methods section, the requested information was added;
The tables have been corrected as indicated;
A conclusion has been added.

We hope that the changes have met your requests.
Thank you very much

Reviewer 2 Report

I appreciate the opportunity to review :  How is leisure related to wellbeing and to substance use? The 2 probable key role of autonomy and supervision 

The paper is concentrated  on motivation as  psychological empowerment.  

Excellent : Participants were 8215 Portuguese adolescents, randomly chosen from those attending the 6th, 8th, 10th and 12th grades in 2018.   

- The present research is  based on a large and representative national survey (line 34) 

Congratulation :  Tables ,...  1. Measures and variables under study....

The authors processed the data very precisely.  

Some problems: 

please correct reference n. 35 

HOEBEN EM, WEERMAN FM. WHY IS INVOLVEMENT IN UNSTRUCTURED SOCIALIZING RELATED TO ADOLESCENT 368 DELINQUENCY?*. Criminology. 2016 May;54(2):242–81. 

references n. 17 and n. 23 - add english translation (check all of them)  

Problem - missing Conclusion - add min. 2 paragraphs  Conditio sine qua non add. 

It would be good implement to Introduction -  (think about it) 

MaturkaniÄŤ, P.; Tomanová ÄŚergeĹĄová, I.; KoneÄŤná, I.; Thurzo, V.; Akimjak, A.; Hlad, Ä˝.; Zimny, J.; Roubalová, M.; Kurilenko, V.; Toman, M.; PetrikoviÄŤ, J.; PetrikoviÄŤová, I also appreciate the overall structure of the article and that it was written with interest and very consistentlyL. Well-Being in the Context of COVID-19 and Quality of Life in Czechia. Int. J. Environ. Res. Public Health 202219, 7164. https://doi.org/10.3390/ijerph19127164

Murgaš, F., PetroviÄŤ, F., MaturkaniÄŤ, P., & Kralik, R. (2022). Happiness or Quality of Life? Or Both?. Journal of Education Culture and Society13(1), 17–36. https://doi.org/10.15503/jecs2022.1.17.36

https://jecs.pl/index.php/jecs/article/view/1408 

I also appreciate the overall structure of the article and that it was written with interest and very consistently. (except for the missing conclusion)  

(Discussion is for authors - conclusion ... anyway - divide...) 

Author Response

Dear reviewer,
Thank you very much for your comments, which help us to improve our article.
We try to make all suggested changes and corrections, as you can see in the submitted article.

A conclusion has been added;

 Reference 35 has been fixed;

Reference no. 17 and 23 English version added;

We hope that the changes have met what you expected.

The introduction has been subject to some changes.

Thank you.

Reviewer 3 Report

Dear Authors, 

Please review the document.

Thanks. 

Dear authors,

It is a study that can have an impact on public policies and on the individual initiative of parents towards their adolescent children, from the association between the variables of interest (leisure-time activities, well-being, substance use). It is hoped that it can also be projected in the research field, probably with younger age ranges (primary school, pre-school). For its part, it has been indebted to enrich the discussion of the investigation.

Some considerations

1. Obsolescence of the data (2018)? (line 17).

2. Indicate if the 42 schools were public, private or mixed. It must be characterized (line 152).

3. Why do you say that it is representative? Did you use expansion factors from the national survey? Was the sample weighted at the national level? They must justify this claim. (line 158).

4. Will it be suitable? Is this questionnaire obsolete (Cantril, 1965)? (line 170).

5. It remains to indicate the level of significance chosen, other statistics for the descriptive analysis (mean, SD, normality...). The statistical analysis section is very short, so it should be increased in content (line 176).

6. The main findings of the study should be highlighted with more emphasis and more clearly. These should go in the first paragraph of the discussion (249 – 251).

7. These ideas are important and must be supported by bibliographical sources (line 263 – 268).

8. The conclusions of the study are not noticed, they have not been emphasized. There are no strengths or weaknesses in the research. They must be done. Finally, the discussion with the comparative evidence is very scarce, this section should be increased in content (line 284).

Thanks.

Author Response

Dear reviewer,
Thank you very much for your valuable comments, which helped us to improve our paper.
They follow answers to their questions, hoping to meet what they expected.

  1. The 2018 data is what we are authorized to work with. There are already data for 2022, but they are still being cleaned, in order to begin their in-depth analysis. As you must understand, the HBSC has a lot of data and sometimes the 4 year interval is not enough to analyze all of them.
  2. Information has been added to the text.
  3. The sample was weighted at the national level. Information has been added to the text.
  4. The Cantril is a very important baseline reference for the type of questionnaire described for this item, which is why we always use the reference.
  5. The questions used are described in the appropriate table, they are isolated questions, not allowing further analysis than those presented. Added information about the scale used.
  6. Information has been added to the text.
  7. Sorry, as we had already made changes to the text, we were unable to identify the referred lines
  8. An article conclusion has been added.

Thank you.

Round 2

Reviewer 1 Report

Dear Authors,

Thank you for the improved version of your manuscript.

Author Response

Dear Reviewer,

Thank you.

Reviewer 3 Report

The authors did not fully respond to corrections. That is, bibliographic support, strengths and weaknesses, and the discussion section does not increase in content. Specifically:

"7. These ideas are important and must be supported by bibliographical sources (line 263 - 268)."

"8. ... There are no strengths or weaknesses in the research. Finally, the discussion with the comparative evidence is very scarce, the content of this section should be increased (line 284)".

Thanks.

Author Response

Dear reviewer,

Thank you for your suggestions.

7. These ideas are important and must be supported by bibliographical sources (line 263 - 268)."

Sorry, I don't know if we identified these lines, if they are the lines of the results, they are sustained in the discussion.

"8. ... There are no strengths or weaknesses in the research. Finally, the discussion with the comparative evidence is very scarce, the content of this section should be increased (line 284)".

We add the limitations of the study, thus supporting its weaknesses and try to improve the discussion.

We hope it meets your expectations.